# CAN STANDARD TRAINING WITH CLEAN IMAGES OUT-PERFORM ADVERSARIAL ONE IN ROBUST ACCURACY?

## ABSTRACT

The deep learning network has achieved great success in almost every field. Unfortunately, it is very vulnerable to adversarial attacks. A lot of researchers have devoted themselves to making the network robust. The most effective one is adversarial training, where malicious examples are generated and fed to train the network. However, this will incur a big computation load. In this work, we ask: "Can standard training with clean images outperform adversarial one in robust accuracy?" Surprisingly, the answer is YES. This success stems from two innovations. The first is a novel loss function that combines the traditional cross-entropy with the feature smoothing loss that encourages the features in an intermediate layer to be uniform. The collaboration between these terms sets up the grounds for our second innovation, namely Active Defense. When a clean or adversarial image feeds into the network, the defender first adds some random noise, then induces this example to a new smoother one via promotion of feature smoothing. At that point, it can be classified correctly with high probability. Thus the perturbations carefully generated by the attacker can be diminished. While there is an inevitable clean accuracy drop, it is still comparable with others. The great benefit is the robust accuracy outperforms most of the existing methods and is quite resilient to the increase of perturbation budget. Moreover, adaptive attackers also fail to generate effective adversarial examples as the induced perturbations overweight the initial ones imposed by an adversary.

## 1 INTRODUCTION

The seminal work of (Goodfellow et al., 2015) pointed out a surprising weakness of modern deep neural networks: although they can perform on par with human beings, their reliability is far from satisfaction. Almost imperceptibly added perturbations will be enough to mislead the network to output a wrong class label with high confidence. It will dramatically undermine the deployment of networks in some safety-critical applications: autonomous driving, image-based ID verification, and medical image analysis.

Since then, researchers have heavily investigated this risk exposure and proposed different defense strategies. One direction is some prepossessing techniques such as bit-depth reduction (Xu et al., 2018), JPEG compression, total variance minimization, image quilting (Guo et al., 2018), and Defense-GAN (Samangouei et al., 2018). The idea is to mitigate the effect of added noise and save the network to some extent. Unfortunately, (Athalye et al., 2018) showed that most of these approaches are based on obfuscated gradients and can be defeated.

The other line of research adopts various adversarial training techniques where malicious examples are generated and fed to the network. A simple rationale behind this is if the network has this knowledge, it will become wise in test time. While there are different mechanisms such as Mixup inference (Pang et al., 2020), feature scattering (Zhang & Wang, 2019), feature denoising (Xie et al., 2019), geometry-aware instance reweighting (Zhang et al., 2021), and channel-wise activation suppressing (Bai et al., 2021), they all share the same philosophy.

While people are astonished by the fact that imperceptibly added perturbations can fool the network, some theoretical works such as (Tsipras et al., 2019; Schmidt et al., 2018) showed that it is not entirely unexpected. Unfortunately, there are no solutions without the awareness of attack models.

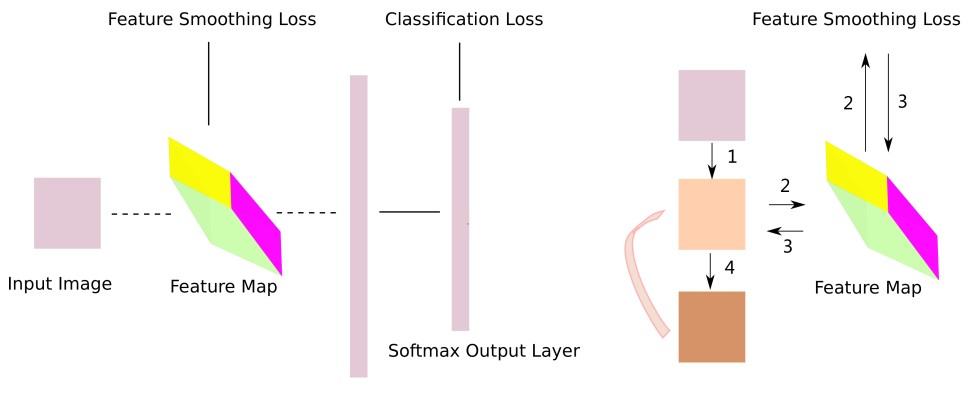

Figure 1: Schematic diagram of our approach. On the left is our standard training stage. The input image undergoes a series of deep neural network units, for example, convolution, ReLU, batch normalization, etc., until the final softmax layer outputs the decision probability. Each input image will produce corresponding feature maps. In addition to the usual classification loss, we introduce the feature smoothing loss to constrain the intermediate feature space. On the right is our Active Defense stage. In total, there are four steps. In Step 1, some random noise is injected into the input. Then this noisy image feeds into the network. During the forward pass (Step 2), we get the feature smoothing loss, and the derivative to the input in the backward pass (Step 3) will be adopted to smooth this noisy image (Step 4). This smoothing process (Steps 2, 3, and 4) can iterate many times until it meets some criterion, then the final result is fed into the network for classification.

Ideally, all defenses should be ignorant of this. However, this knowledge is essential to the adversarial training method that remains most effective, although at the cost of a large computation load. Now the big question arises: **"Can standard training with clean images outperform adversarial one in robust accuracy?"** Here "clean images" means there is no manipulation of inputs even by adding some random noise such as (Jin & Rinard, 2020), although it is for manifold regularization rather than adversarial training. At first glance, it seems hopeless, as a widely accepted principle in the adversarial learning community is that a network can be clever only if it has been exposed to deceptions before. However, on the other hand, the networks are supposed to generalize well after standard training. How can it perform so badly for adversarial attacks? As a possible answer to this, (Ilyas et al., 2019) investigated the cause of adversarial examples and concluded that neural networks tend to exploit predictive yet brittle features to make classifications. These features are incomprehensible to humans and thus can be modified by adversarial attackers to mislead the networks, but (Ilyas et al., 2019) did not show how to teach the network to disregard these non-robust features and discover the robust ones to make final decisions. From this perspective, as it is difficult to tell the network to learn robust features, what if we add some hints in the loss function and let the network become robust in an implicit way? More specifically, in addition to the classical cross-entropy loss, we use a feature smoothing term that encourages the features in an intermediate layer to be uniform, as shown in the left of Figure 1. It sounds counterintuitive as this term will constrain the space of features that may lead to a wrong classification. However, due to the high capacity of networks, a very high standard accuracy can still be achieved with this additional term.

When training completes, given an input, whether clean or crafted, extra perturbations can always be created by some added random noise followed by the promotion of feature smoothing at the cost of reduced accuracy. So long as these intentional perturbations overweight the adversary's and the reduction in accuracy is affordable, the model will become robust. We call this procedure Active Defense, as shown in the right of Figure 1. We find experimentally, a clean example from CIFAR-10/CIFAR-100 can be perturbed with $l_\infty = 25/255 \sim 32/255$, three to four times of $l_\infty = 8/255$ usually adopted by an adversary, yet be classified with a high success rate. This fact sets up the adequate space for Active Defense for eliminating the effects of attacks. Our approach is independent of any attack models compared with other state-of-the-art methods, and its performance is much

more stable under attacks with different budgets. The contributions of this work are summarized as follows:

• We propose a novel training scheme with an extra feature smoothing loss term that only takes clean images as inputs, fundamentally different from all existing adversarial training methods that need supplementary crafted data.

• We present Active Defense that adds the second round of perturbations through random noise and feature smoothing. It modifies the malicious examples in a way that is friendly to the network. This deviates from conventional passive ones that keep the input intact.

## 2 RELATED WORKS

Due to adversarial threats to deep learning applications, there are many works to improve the robustness. Most of them adopt adversarial training. Among them, only a few take care of the features in intermediate layers as listed in the following.

Feature denoising in (Xie et al., 2019) found that small perturbations in pixel space can lead to very substantial noise in the feature maps of the network and proposed various filters to denoise. (Zhang & Wang, 2019) proposed to generate adversarial images for training through feature scattering in the latent space. In essence, perturbed images are produced collaboratively via optimal transport distance minimization. (Zhang & Wang, 2019) used the feature maps as a guide to making new examples. Compared with these two, we are trying to force the intermediate feature map to be uniform through an additional loss term within the standard training framework without any modification of the network as in (Xie et al., 2019) or any other manipulations of features as in (Zhang & Wang, 2019).

Regarding Active Defense, we have not seen any similar work. Perhaps the most related one is (Yang et al., 2019), which used sophisticated matrix completion techniques to reconstruct the random masked images. Our motivation is very different, as we try to exploit the deep network itself to enhance robustness without borrowing any third-party algorithms.

## 3 BACKGROUND

In the classification problem, given the training data set of image-label pairs $D = \{(x_i, y_i)\}_{i=1}^n$ where $y_i \in \{1, 2, ..., M\}$, the goal is to find an output probability vector $F(x)$ of length $M$ indexed by $j$, ideally such that $y = \arg\max_j F_j(x)$. Of course, there is always a mismatch between these two terms. The key thing here is to find a suitable loss function such that the empirical risk minimization (ERM) of $\frac{1}{n} \sum_{i=1}^n L(F(x_i), y_i)$ can be implemented with loss function $L$. Note that $F(x_i)$ is a vector, while $y_i$ is a scalar of the label, the very first thing is to transform $y_i$ into a vector through a vector function $G(y_i)$. People usually adopt the one-hot coding $H(y_i)$ of length $M$ with all elements being 0 except $H_{y_i}(y_i) = 1$. The two probability distribution vectors $F(x_i)$ and $H(y_i)$ can be compared with cross-entropy.

An adversary crafts an adversarial example $x_{adv}$ which is closest to $x$ with $\|x_{adv} - x\|_p \le \varepsilon$ but misclassified as some other class. In this paper, we only consider attacks with $p = \infty$. The most commonly used strategy is the iterative projected gradient descent method(PGD)

$$x_{adv}^{t+1} = P(x_{adv}^t + \beta \times \text{sign}(\nabla_x L(x_{adv}^t, G(y)))), \tag{1}$$

where $\beta$ is the step size and $P$ projects the generated example to the feasible region. Please note that $L$ in adversarial attack may be different from $L$ in training. People may choose traditional cross-entropy loss with one-hot coding or CW loss (Carlini & Wagner, 2017) to implement Equation 1. Currently, a budget-aware step size-free variant of PGD has been proposed by (Croce & Hein, 2020), and since that, an ensemble of diverse parameter-free attacks called AutoAttack has become the de facto routine for robust accuracy evaluation.

## 4 METHOD

In general, our method is very simple. In training, for feature map $F^l$ of a particular layer $l$ with $W \times H \times C$ ,we use the loss

$$L = L_{ce} + max\left(L_{F^l}, \delta\right) \tag{2}$$

$$L_{F^l} = \frac{1}{W \times H \times C} \sum_{i=1}^{W} \sum_{j=1}^{H} \sum_{k=1}^{C} \left|F^l_{i,j,k} - mean(F^l)\right|. \tag{3}$$

Here $L$ has two terms. $L_{ce}$ is for cross-entropy loss, and $L_{F^l}$ is our novel feature smoothing loss function. It is quite similar to the $L_1$ norm of the particular feature cube and encourages the cube to be uniform. In order to avoid overfitting to the feature smoothing loss, we use $max\left(L_{F^l}, \delta\right)$ which disables the derivative of $L_{F^l}$ when it drops below $\delta$. In summary, we have two parameters, the feature layer $l$ and the smoothing upper bound $\delta$. Although this loss function sounds straightforward for our purpose to make the intermediate feature map smooth, there is a novel insight from the perspective of the trade-off between these two terms. In order to get low $L_{F^l}$, $L_{ce}$ will increase. In other words, after training, the network somehow understands that feature smoothing is not very annoying, and it only causes accuracy to drop to some extent. There is a huge implication in terms of robust accuracy. If we only use $L_{ce}$, the network has no idea to deal with the crafted example except to be fooled. However, in our case, $L_{F^l}$ gives us a dissipation channel of malicious perturbations that we can take via feature smoothing. Hopefully, this will remove most of the perturbations generated by an adversary, which we will elaborate on in our Active Defense design. The other concern may relate to the feature space constraints. Actually, due to the high capacity of networks, there is almost no difference in standard training accuracy between ours and cross-entropy loss.

From the discussion above, our Active Defense is very intuitive. It consists of four steps depicted in the right of Figure 1. In Step 1, we add some random noise to an input which can somehow reduce the effect of adversarial disruptions; more importantly, this will ensure sufficient iterations of feature smoothing. Otherwise, the attacker can bypass this and our Active Defense will fail. Steps 2 and 3 are just forward/backward passes related to $L_{F^l}$. In Step 4, the noisy image get smoothed via gradient descent for $L_{F^l}$, and feeds into the network for another round of feature smoothing. The overall procedure of the proposed approach is in Algorithm 1. There are only three parameters, i.e., $\sigma$ for uniform noise, $\beta$ for the updating step size, and $\tilde{\delta}$ for the upper bound of feature loss in test that is usually lower than $\delta$ for training in Equation 2, as we pursue extra feature smoothing to deal with adversarial attacks. The updated example through Active Defense will feed into the network for final class decision. This algorithm can run a few times which we denoted as $outerloop$ in the following sections.

---

**Algorithm 1** Active Defense Algorithm

---

1: **procedure** ACTIVE DEFENSE$(t, L_{F^l})$              ▷ $t$ is a test example.
2:    $t = t + uniform(-\sigma, \sigma)$     ▷ **Step 1**, $\sigma$ is the parameter of the uniform distribution.
3:    $l = L_{F^l}(t)$                     ▷ **Step 2**
4:    **while** $l > \tilde{\delta}$ **do**         ▷ $\tilde{\delta}$ is the upper bound of feature loss in test.
5:     $d = \frac{\partial l}{\partial t}$                    ▷ **Step 3**
6:     $t = t - \beta \times d$               ▷ **Step 4**, $\beta$ is the step size.
7:     $l = L_{F^l}(t)$                    ▷ **Step 2**
8:    **end while**
9:    **return** $t$
10: **end procedure**

---

## 5 EXPERIMENTS

To evaluate the performance of our approach, we run it on two datasets, CIFAR-10 and CIFAR-100 (Krizhevsky, 2009), and compare it with other state-of-the-art adversarial training methods. The network we choose is WideResNet-28-10 (Zagoruyko & Komodakis, 2016) with three groups. Naturally, feature maps $F^l$ ($l = 0, 1, 2$) are used to denote the outputs of three groups respectively, and we choose $l = 0$, as it is closest to the input with the strongest backward pass derivative. We

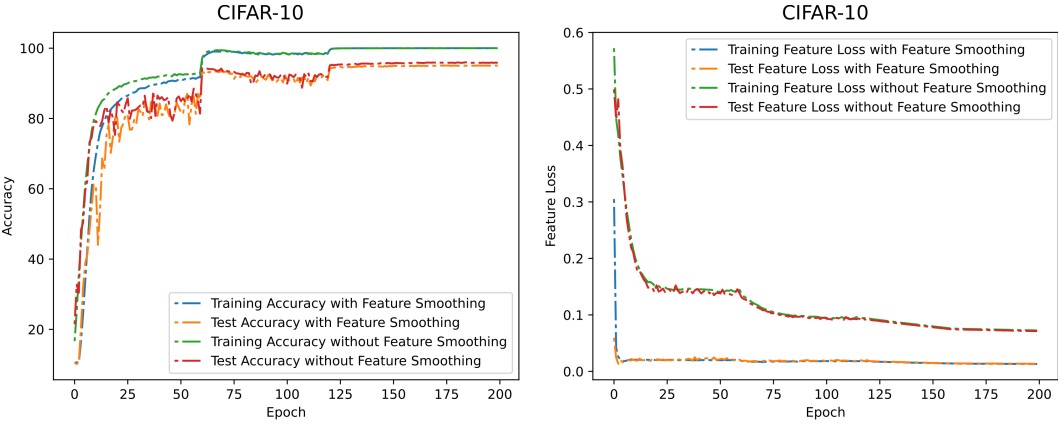

Figure 2: Training/Test accuracy (on the left) and Feature Loss (on the right) for CIFAR-10.

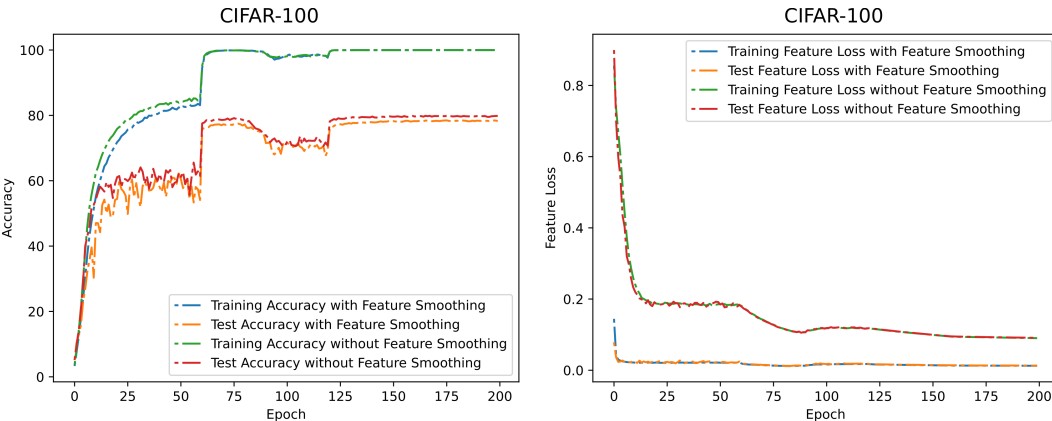

Figure 3: Training/Test accuracy (on the left) and Feature Loss (on the right) for CIFAR-100.

evaluate the robust accuracy with perturbation budget $\varepsilon$ as both $8/255$ and $16/255$. It needs to be emphasized here that our algorithm is independent of $\varepsilon$.

## 5.1 STANDARD TRAINING

Our approach adopts standard training with clean images using an additional feature loss term which boosts the intermediate features to be smooth. In all our experiments, the smoothing upper bound $\delta$ in Equation 2 is set to 0.02. Figures 2 and 3 demonstrate the training and test accuracy and the feature loss variation per training epoch.

As we expected, while they achieve almost 100% training accuracy, there is only a tiny drop in test accuracy with and without feature smoothing terms: 95.02% vs 95.83% for CIFAR-10, and 78.3% vs 79.86% for CIFAR-100. This is due to the fact that the classifiers have a large number of parameters and are powerful enough. Here is an interesting observation of feature loss: even when we do not impose any constraints, it still goes down slowly. While we do enforce this term, it goes down very quickly at the beginning. This strongly hints feature smoothing benefits the generalization ability of networks. The feature loss with and without feature smoothing terms are dramatically different, listed in the pattern of training/test, 0.0130/0.0135 vs 0.0725/0.0713 for CIFAR-10, and 0.01258/0.013 vs 0.0901/0.0909 for CIFAR-100. This fact indicates the effects of this additional loss term. Also, although we set the $\delta$ to be 0.02, the feature loss is lower than that number, especially for CIFAR-100, which is also evidence that classification and feature smoothing are cooperative to some extent. Our Active Defense design exactly takes advantage of this harmony.

Table 1: AA comparison (WideResNet-28-10 only), * denotes models that require additional external images.

| # | paper | clean | $\text{APGD}_{\text{ce}}$ | $\text{APGD}_{\text{dlr}}^{\text{T}}$ | $\text{FAB}^{\text{T}}$ | Square | AA |
|---|---|---|---|---|---|---|---|
| **CIFAR-10** - $l_\infty$ - $\varepsilon = 8/255$ | | | | | | | |
| 1 | Ours ($outerloop = 10$) | 86.69 | 73.71 | 77.64 | 85.30 | 79.03 | 68.54 |
| 2 | Gowal et al. (2020)* | 89.48 | 65.64 | 62.83 | 63.29 | 68.74 | 62.76 |
| 3 | Ours ($outerloop = 1$) | 85.35 | 73.10 | 74.11 | 83.49 | 76.73 | 62.31 |
| 4 | Rade & Moosavi-Dezfooli (2021) | 88.16 | 63.75 | 60.99 | 61.39 | 66.67 | 60.97 |
| 5 | Rebuffi et al. (2021) | 87.33 | 63.99 | 60.76 | 61.14 | 66.52 | 60.73 |
| - | Standard (using Active Defense) | 16.20 | 16.13 | 15.94 | 16.10 | 15.79 | 7.40 |
| **CIFAR-100** - $l_\infty$ - $\varepsilon = 8/255$ | | | | | | | |
| 1 | Ours ($outerloop = 10$) | 59.43 | 39.36 | 44.78 | 57.63 | 48.92 | 33.49 |
| 2 | Rebuffi et al. (2021) | 62.41 | 35.66 | 32.08 | 32.30 | 36.76 | 32.06 |
| 3 | Ours ($outerloop = 1$) | 56.98 | 37.90 | 42.85 | 55.04 | 46.80 | 28.51 |
| 4 | Hendrycks et al. (2019)* | 59.23 | 33.02 | 28.48 | 28.74 | 34.26 | 28.42 |
| 5 | Rice et al. (2020) | 53.83 | 20.57 | 18.98 | 19.24 | 23.57 | 18.95 |
| - | Standard (using Active Defense) | 2.69 | 2.54 | 2.65 | 2.78 | 2.56 | 1.39 |

## 5.2 ACTIVE DEFENSE

Equation 2 takes two terms, $L_{F^l}$ and $L_{ce}$. What is the consequence if we make the image smoother further, i.e., to decrease $L_{F^l}$? This essentially will remove some brittle features, however, since we train the network with $L_{F^l}$, the network somehow understands that the remaining features are still useful for image classification. In other words, the drop in classification accuracy should be small. It is key to our success of Active Defense. To counter adversarial perturbations, we intentionally add some noise and smooth it out as in Algorithm 1. In all our experiments, we choose $\sigma = 0.075$ for uniform noise and $\beta = 80$ for the updating step size, while for the upper bound of feature loss $\tilde{\delta}$, 0.0124 for CIFAR-10 and 0.0118 for CIFAR-100. Some experimental results with $\varepsilon = 8/255$ in Figures 4 and 5 show Active Defense can effectively recover the semantically significant structures destroyed by the adversarial attacks and our intentionally added noise, which leads to correct final classification. This evidence highlights the role played by our Active Defense. Usually, a single Active Defense takes about 200-400 iterations of while loop body in Algorithm 1 for $\varepsilon = 8/255$ and up to around 500 for $\varepsilon = 16/255$. In practice, this while loop body can be implemented in parallel.

We evaluate the robust accuracy of our approach using AutoAttack and compare the performance with other state-of-the-art methods in Tables 1 and 2. As AutoAttack comprises a set of attack methods, it will win if any of them succeeds. It poses some challenges for our approach since our solution is stochastic. As one attack model may not be so strong on an average of 10k images, but for a particular example, it still has some chance to win. This chance will increase especially when given a row of ones applied in sequence, namely, clean, $\text{APGD}_{\text{ce}}$, $\text{APGD}_{\text{dlr}}^{\text{T}}$, $\text{FAB}^{\text{T}}$ and Square. For example, in Table 1, $outerloop = 1$ for CIFAR-10 achieves the lowest accuracy of 73.10% for $\text{APGD}_{\text{ce}}$, but AA only gets 62.31%, more than a ten percent drop. However for other comparison methods, this gap is small. To further improve the stability and narrow this gap, we run it ten times, i.e., $outerloop = 10$, and aggregate the output probability of each run.

In Table 1, we only consider WideResNet-28-10, and ours with $outerloop = 10$ is the best. One may still wonder what if we apply our Active Defense with classical training only with cross-entropy loss. We also list the results in the rows of Standard (using Active Defense). It turns out that it is hard to exit the while loop in Algorithm 1, so we set $\tilde{\delta}$ to be 0.0562 for CIFAR-10 and 0.0880 for CIFAR-100. The accuracy is very low. This fact justifies the necessity of our feature smoothing loss term. Table 2 lists the AA accuracy with all model architectures, and ours is still among the best. Note that when $\varepsilon = 16/255$, we rerun the whole AA including clean with a little bit change in clean accuracy due to the randomness of our method, and our AA accuracy significantly outperforms all others.

Table 2: AA comparison, * denotes models that require additional external images.

| # | paper | architecture | clean | AA |
|---|-------|-------------|-------|-----|
| **CIFAR-10** - $l_\infty$ - $\varepsilon = 8/255$ | | | | |
| 1 | Ours ($outerloop = 10$) | WideResNet-28-10 | 86.69 | 68.54 |
| 2 | Rebuffi et al. (2021)* | WideResNet-70-16 | 92.23 | 66.56 |
| 3 | Gowal et al. (2020)* | WideResNet-70-16 | 91.10 | 65.87 |
| 4 | Rebuffi et al. (2021) | WideResNet-106-16 | 88.50 | 64.58 |
| 5 | Rebuffi et al. (2021) | WideResNet-70-16 | 88.54 | 64.20 |
| 6 | Rade & Moosavi-Dezfooli (2021)* | WideResNet-34-10 | 91.47 | 62.83 |
| 7 | Gowal et al. (2020)* | WideResNet-28-10 | 89.48 | 62.76 |
| 8 | Ours ($outerloop = 1$) | WideResNet-28-10 | 85.35 | 62.31 |
| 9 | Rade & Moosavi-Dezfooli (2021) | WideResNet-28-10 | 88.16 | 60.97 |
| 10 | Rebuffi et al. (2021) | WideResNet-28-10 | 87.33 | 60.73 |
| **CIFAR-10** - $l_\infty$ - $\varepsilon = 16/255$ | | | | |
| 1 | Ours ($outerloop = 10$) | WideResNet-28-10 | 86.79 | 54.24 |
| 2 | Ours ($outerloop = 1$) | WideResNet-28-10 | 85.21 | 49.24 |
| 3 | Rebuffi et al. (2021) | WideResNet-28-10 | 87.33 | 25.39 |
| 4 | Gowal et al. (2020)* | WideResNet-28-10 | 89.48 | 24.96 |
| 5 | Wu et al. (2020)* | WideResNet-28-10 | 88.25 | 24.00 |
| **CIFAR-100** - $l_\infty$ - $\varepsilon = 8/255$ | | | | |
| 1 | Gowal et al. (2020)* | WideResNet-70-16 | 69.15 | 36.88 |
| 2 | Rebuffi et al. (2021) | WideResNet-70-16 | 63.56 | 34.64 |
| 3 | Ours ($outerloop = 10$) | WideResNet-28-10 | 59.43 | 33.49 |
| 4 | Rebuffi et al. (2021) | WideResNet-28-10 | 62.41 | 32.06 |
| 5 | Cui et al. (2020) | WideResNet-34-20 | 62.55 | 30.20 |
| 6 | Gowal et al. (2020) | WideResNet-70-16 | 60.86 | 30.03 |
| 7 | Cui et al. (2020) | WideResNet-34-10 | 60.64 | 29.33 |
| 8 | Rade & Moosavi-Dezfooli (2021) | PreActResNet-18 | 61.50 | 28.88 |
| 9 | Wu et al. (2020) | WideResNet-34-10 | 60.38 | 28.86 |
| 10 | Ours ($outerloop = 1$) | WideResNet-28-10 | 56.98 | 28.51 |
| 11 | Rebuffi et al. (2021) | PreActResNet-18 | 56.87 | 28.50 |
| **CIFAR-100** - $l_\infty$ - $\varepsilon = 16/255$ | | | | |
| 1 | Ours ($outerloop = 10$) | WideResNet-28-10 | 59.16 | 22.76 |
| 2 | Ours ($outerloop = 1$) | WideResNet-28-10 | 57.10 | 19.15 |
| 3 | Rebuffi et al. (2021) | WideResNet-28-10 | 62.41 | 12.45 |
| 4 | Hendrycks et al. (2019)* | WideResNet-28-10 | 59.23 | 8.89 |

## 5.3 ADAPTIVE ADVERSARIES

While it is always not easy to propose adaptive attacks, we try our best to defeat our defense scheme. Since the APGD$_{ce}$ seems to be the best attack as listed in Table 1, we adapted it to generate possibly stronger adversarial examples. Specifically, the loss function of APGD$_{ce}$ is modified to

$$L = L_{ce} + \lambda \times max\left(L_{F^l}, \delta\right). \tag{4}$$

In other words, we try to understand how the feature loss term influences the efficiency of attacks with $\lambda$ spaced uniformly (on a log scale) from 0.1 to 100, both positive and negative. Positive $\lambda$ means more distortion in feature space, which pressures our smoothing process via Active Defense. While for negative $\lambda$, the iterations of the while loop in Algorithm 1 should be small, which results in more effective attacks. However these efforts are useless, manifested in robust accuracy of Table 3, almost the same as $\lambda = 0$.

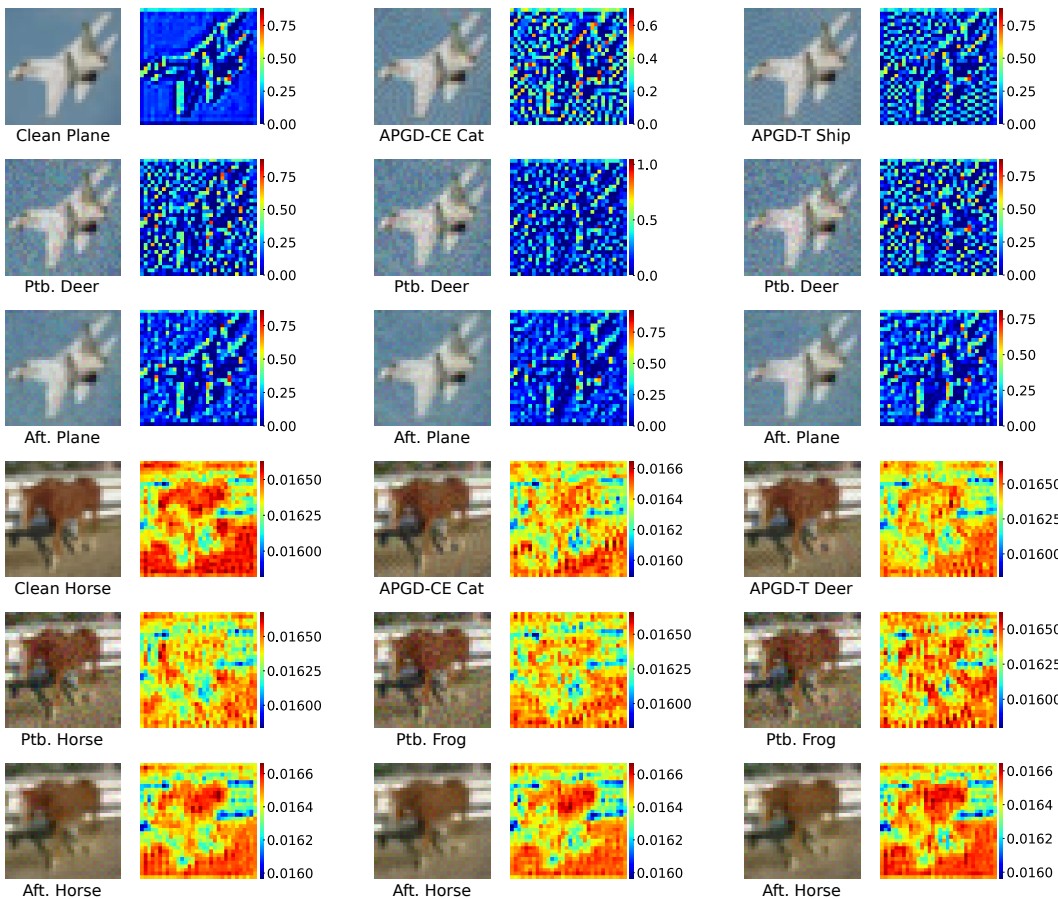

Figure 4: The top half is a plane from CIFAR-10. In the first row, from left to right, we demonstrate the clean and adversarial images generated via $\text{APGD}_{\text{ce}}$ and $\text{APGD}_{\text{dlr}}^{\text{T}}$, and the corresponding feature map in channel 7. The second and third rows are the noise-perturbed and after Active Defense ones, respectively. The second half is arranged similarly except that the feature channel is 26. Our standard training model assigns each image a label shown underneath it. The feature maps show strong distortions by adversarial attacks and our intentionally added noise, which can be somehow restored by our Active Defense.

Table 3: Robust accuracy on adaptive attacks with different $\lambda$ through a single run of Active Defense. The left half table is for negative $\lambda$, while the remaining one is for positive $\lambda$ in bold.

| $\log_{10}(|\lambda|)$ | -1 | -0.25 | 0.5 | 1.25 | 2 | **-1** | **-0.25** | **0.5** | **1.25** | **2** |
|---|---|---|---|---|---|---|---|---|---|---|
| **CIFAR-10** | 72.95 | 73.06 | 72.62 | 72.95 | 72.79 | **72.87** | **72.91** | **72.7** | **72.76** | **72.43** |
| **CIFAR-100** | 37.72 | 38.06 | 38.02 | 37.32 | 37.72 | **37.79** | **37.69** | **37.66** | **37.64** | **37.32** |

## 5.4 DISCUSSION

All above attacks are one-time in nature, as attackers may take an arbitrarily long time and possibly get many options but can only shoot one. Since our method introduces random noise in Active Defense phase, the results demonstrate the average success rate for 10K test examples. One may come up with a simple attack that sends the same example many times, and there is always a chance to defeat. But this brute-force attack can be resolved by our enhanced version of defense, that is, to run a lot of times of Algorithm 1, and aggregate the probability of each run. However, this will increase the computation load. On the other hand, it makes sense since there is no free lunch. Also, very fortunately, our capacity of defense can scale up much more conveniently than perhaps retraining the model from scratch as conventional adversarial training methods. Another very nice

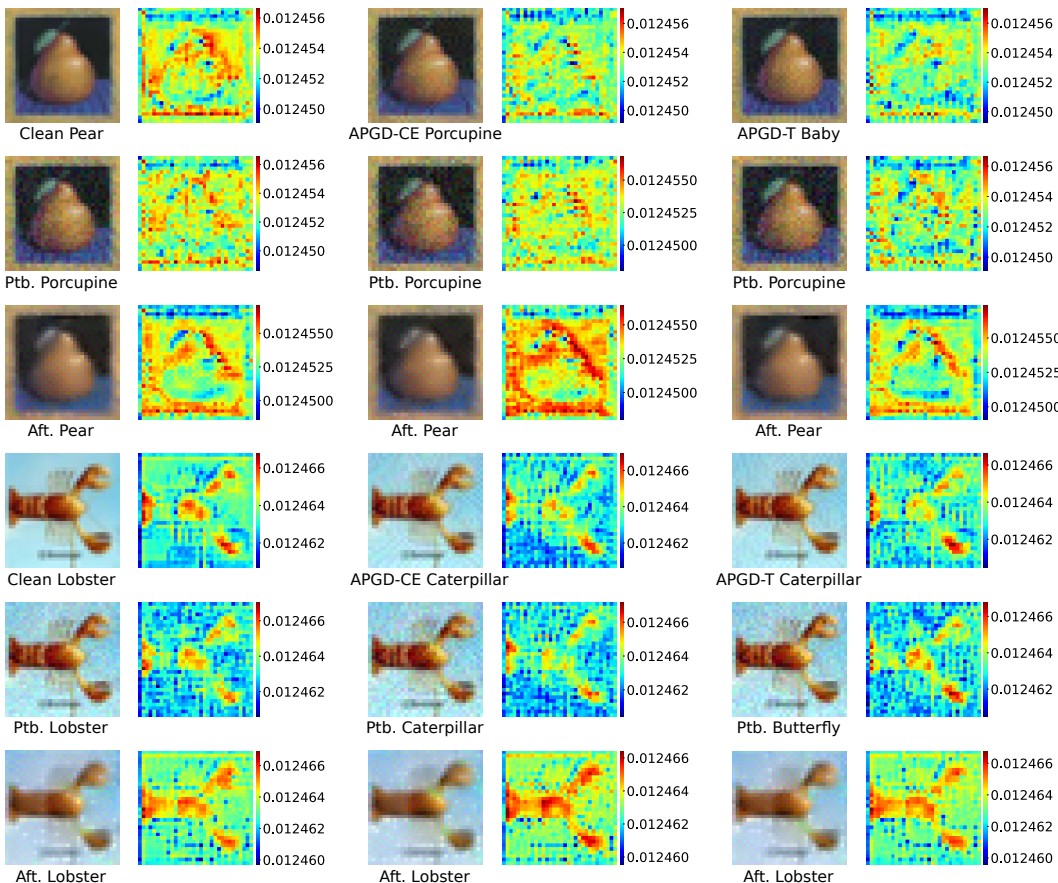

Figure 5: The top half is a pear from CIFAR-100. In the first row, from left to right, we demonstrate the clean and adversarial images generated via $\text{APGD}_{\text{ce}}$ and $\text{APGD}_{\text{dlr}}^{\text{T}}$, and the corresponding feature map in channel 43. The second and third rows are the noise-perturbed and after Active Defense ones, respectively. The second half is arranged similarly except that the feature channel is 33. Our standard training model assigns each image a label shown underneath it. We can see the similar destructions and restorations as in Figures 4. Moreover, comparing with the clean image, the lobsters in feature maps are much more integrated in the last row.

advantage is that our enhancement is definitively free from robust overfitting (Rice et al., 2020), as there is no attack model engaged at all.

## 6 CONCLUSION

Adversarial learning is of great interest to the deep learning community. Most of the previous works focus on the efficient generation of malicious examples. However, in this paper, we shed some light on a question: Is it possible that a network can be robust without being taught with malicious examples? We propose a standard training model with an additional feature smoothing loss term, which is very different from all existing ones in that there is no adversarial input involved. The standard cross-entropy and feature smoothing loss can collaborate to some extent in training. At test time, we adopt Active Defense to distill feature maps of adversarial inputs. The experimental results demonstrate that this simple method can enhance the robustness of networks greatly. In future work, we will do some theoretical analysis and exploit other forms of cooperative loss that might be more beneficial than the feature smoothing one.

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
