# OpenReview forum: "Can standard training with clean images outperform adversarial one in robust accuracy?"
_ICLR.cc/2022/Conference — ICLR 2022 Submitted_

### Official Review · Reviewer_tMiw · 2021-10-17

**Correctness:** 2
**Technical Novelty And Significance:** 2
**Empirical Novelty And Significance:** 2
**Recommendation:** 3
**Confidence:** 4

**Main Review:**

The most significant concern I have is that the “adaptive attack” is not, in my view, a sufficient try at breaking the proposed method.  The authors should perform an adaptive PGD attack where each gradient is computed as follows: Draw multiple samples of random noise, add each of them individually to your current adversarial example iterate, run Algorithm 1 for each of the noised examples, plug the resulting images into the model to obtain a cross-entropy (also try CW) loss value, and finally backprop through the entire Algorithm 1 to compute a gradient with respect to the adversarial example iterate.

Imagenet experiments would be greatly appreciated as ImageNet data is far higher dimensional, more realistic, and typically has very different properties than low-dimensional data.

There are several issues with the writing of Algorithm 1:  What is lower case ell?  Also, s is only computed and never used or output.

Just a note about formatting.  The authors should be careful about the difference between \citep and \citet natbib commands.  Additionally, there are numerous grammatical errors and misspellings which should be corrected before publication.

**Summary Of The Paper:**

The paper proposes a new defense strategy involving both randomization and smoothing.  To my knowledge, no other paper has proposed precisely the same defense.

**Summary Of The Review:**

The "adaptive attack" is not sufficient and needs improvement.  I am highly skeptical that this defense will stand up to scrutiny, and I don't think the current experiments demonstrate that it will.

---

> ### Author Response · Authors · 2021-11-12
> **Response to Reviewer tMiw**
>
> “The most significant concern I have is that the “adaptive attack” is not, in my view, a sufficient try at breaking the proposed method.”
>
> Sorry but No, this attack won’t work. Let’s suppose the input is X, after being added to by noise, we get X1, X2. Since the noise is so high, with sigma=19/255, X1, X2, and X are very different for each pixel. We can’t expect that any hint for X1/X2 to be manipulated as an adversarial sample could be used as a guide for X.
>
> Moreover, the crafted image still needs to undergo another noise injection, which is totally independent of previous ones, the above effort will turn out to be useless.
>
> “Imagenet experiments would be greatly appreciated”
>
> Yes we are also excited to see the results, however due to limited facilities, we can’t try that right now.
>
> "There are several issues with the writing of Algorithm 1: What is lower case ell? Also, s is only computed and never used or output."
>
> Sorry about that. In Alg 1, “s” should be “l”.
>
> "Just a note about formatting. ..."
>
> Thank you very much. Definitely, we will correct them.

---

> > ### Comment · Reviewer_tMiw · 2021-11-12
> > **This does not address my concern over adaptivity.**
> >
> > I completely understand how your "active defense stage" works, and your rebuttal does not address my concerns over the "adaptive" attack.  The goal of an adversary is to maximize the expected loss incurred by the defense strategy where the expectation is taken over all random components of the defense.  The technique of estimating this expected loss by sampling over randomizations is known as Expectation Over Transformation (EOT) and is used in numerous seminal adversarial attack works including "Synthesizing Robust Adversarial Examples", "On Adaptive Attacks to Adversarial Example Defenses", "On Evaluating Adversarial Robustness", "Obfuscated Gradients Give a False Sense of Security: Circumventing Defenses to Adversarial Examples", etc.  These works discuss how defenses with randomness are not actually more secure. By sampling many noise instances, unrolling the defense, and averaging the gradients w.r.t. inputs over all of the noise instances, an adaptive adversary can bypass the defense.  The "adaptive" attack in your paper is not a true adaptive attack, and I do not believe that your defense is truly robust until I see results on a truly adaptive attack.

---

### Official Review · Reviewer_R7ZE · 2021-10-26

**Correctness:** 2
**Technical Novelty And Significance:** 2
**Empirical Novelty And Significance:** 2
**Recommendation:** 3
**Confidence:** 3

**Main Review:**

The motivation of its current version is unclear. The performance gain of robustness is not convincing.

- The feature smoothing method is proposed for training robust model, but it also is used in the test phase. The motivation for applying this strategy in the test phase is unclear. Intuitively, removing the inference-time smoothing operation (the backpropagation process) will degrade the accuracy of the proposed method.
- The proposed method is actually ‘attack’ the generation process of adversarial examples, so this strategy may give false robustness. This kind of approach is not novel [1].
- Besides adaptive attack, the criterion suggested by [2] is also necessary to verify the robustness.

[1] Fighting Gradients with Gradients: Dynamic Defenses against Adversarial Attacks. arXiv
[2] Obfuscated gradients give a false sense of security: Circumventing defenses to adversarial examples. ICML2018


**Summary Of The Paper:**

This paper aims to provide an approach to train a robust model, where the model is trained without adversarial examples. To this end, the authors introduce an inference-time defense strategy. However, this idea is not novel. Moreover, the inference-time defense strategy may give false robustness.

**Summary Of The Review:**

The motivation of its current version is unclear. The performance gain of robustness is not convincing.

---

> ### Author Response · Authors · 2021-11-12
> **Response to Reviewer R7ZE**
>
> “The feature smoothing method is proposed for training robust model, but it also is used in the test phase. ...”
>
> Please refer to the common comment 2.
>
> “The proposed method is actually ‘attack’ the generation process of adversarial examples, so this strategy may give false robustness. This kind of approach is not novel [1].”
>
> Thank you for bringing us the awareness of [1].
> [1] is totally different from ours. They have to change the network parameters while ours is fixed. So their model may drift. Moreover, the parameter to be adapted are related to the batch normalization layer, so it is dependent on batch size. Ours is free from all these problems.
> The most prominent is that they only consider epsilon 1.5/255 for the standard training model with 45.4% accuracy, which is far lower than our 68.54% for epsilon 8/255.
>
> Can you please give more specific reasons why our model may give false robustness?
>
> "Besides adaptive attack, the criterion suggested by [2] is also necessary to verify the robustness."
>
> Please refer to the common comment 3.

---

> > ### Comment · Reviewer_R7ZE · 2021-11-27
> > **Thank you for your response**
> >
> > I thank the authors for addressing my questions, but I am prone to keep my rating, considering that the mentioned two points are necessary to verify the proposed method. That is,
> > - randomized defenses
> > - optimization at inference time.
> >
> > Thus, I suggest the authors conduct the adaptive attack to validate the proposed method further.

---

### Official Review · Reviewer_qx2w · 2021-11-02

**Correctness:** 4
**Technical Novelty And Significance:** 3
**Empirical Novelty And Significance:** 4
**Recommendation:** 5
**Confidence:** 4

**Main Review:**

Strengths
1. The paper develops an effective and simple adversarial defense that requires only adding a simple regularization term during training, and additional processing of inputs during testing.
2. The proposed defense appears to outperform baseline defenses on strong attacks. The accuracies achieved by the defense appear competitive.

Weaknesses
1. The proposed defense is novel, but some ideas are similar to ones previously proposed in the literature. In particular, the idea of feature smoothing is related to prior work as noted in section 2. Moreover, the idea of adding noise to inputs as a defense has been explored in the literature on randomized smoothing.
2. The proposed defense is not theoretically motivated. This is not strictly necessary given the strong empirical results; however, it would be helpful to provide further intuition on why the interaction between feature smoothing during training and during active defense may enhance adversarial robustness.
3. It is not clear to what extent the noise (step 1 of active defense) by itself enhances robustness vs. the feature smoothing step of the active defense. The authors demonstrate that standard training by itself with active defense is ineffective, indicating the importance of the feature smoothing loss during training. It would be helpful to conduct a similar ablation experiment to evaluate the importance of the noise vs. feature smoothing steps in the active defense.
4. It is possible that the high performance of the proposed method is due to obfuscated gradients since for the attacker, optimizing inputs through the active defense may be more difficult. Adding additional experiments to demonstrate that obfuscated gradients do not occur would strengthen the paper (the authors may want to consider the experiments in Athalye et al., 2018).
5. The authors do not state whether the proposed method is state-of-the-art. To the best of their knowledge, do the authors achieve state-of-the-art robust accuracy on CIFAR-10 at $\epsilon=8/255$?
6. What is the role of the L1/absolute value penalty in eqn 3? Would the method still be effective with an L2/square penalty?

Minor comments
1. Some typos: "AutoAttak" in section 3, " It needs to emphasize here " -> " It needs to be emphasized here" in section 5



**Summary Of The Paper:**

This paper proposes a defense against adversarial attacks that does not involve using attacks in the defense process. The defense consists of two parts: first, a feature smoothing loss is added to the main objective function during training. Then, when defending against attacks, noise is added to the input followed by additional perturbations to the input designed to decrease the feature smoothing loss. The proposed method achieves favorable accuracies relative to baseline defenses.

**Summary Of The Review:**

The proposed method appears to achieve state-of-the-art performance. However, fully demonstrating this will require addressing the possibility of obfuscated gradients. Moreover, understanding the role of noise in the active defense via an ablation experiment will be important to see which aspects of the method actually improve robustness. If the authors can address these points, the paper would be significantly improved.

---

> ### Author Response · Authors · 2021-11-12
> **Response to Reviewer qx2w**
>
>
> 1.The proposed defense is novel, but some ideas are similar to ones previously proposed in the literature
>
> Ours is very different from all existing ones in that we don’t need specially crafted images. In randomized smoothing, they train the base classifier with Gaussian data augmentation. In (Yang et al., 2019), they also need random masked images for training.
>
> 2.The proposed defense is not theoretically motivated
>
> Please refer to the common comment 2 for some intuitive interpretations.
> Theoretical analysis is our future work.
>
> 3.It is not clear to what extent the noise (step 1 of active defense) by itself enhances robustness.
>
> This step is essential. The exit condition in the while loop is based on the smoothing loss. A sufficient noise, in our case 19/255, is necessary to ensure the enough iterations of feature smoothing within the while loop. Otherwise, an adversary can easily craft images that can bypass this feature smoothing process, our defense is then broken. As a matter of fact, we are always bearing in mind the possible adaptive attacks when we design our defense.
>
> 4.It is possible that the high performance of the proposed method is due to obfuscated gradients…
>
> Please refer to the common comment 3.
>
> 5.The authors do not state whether the proposed method is state-of-the-art.
>
> Sure, it is state-of-the-art.
>
> 6.What is the role of the L1/absolute value penalty in eqn 3? Would the method still be effective with an L2/square penalty?
>
> Yes, it is possible to try another penalty, so long as it can promote feature uniformity.
>
> 7.Some typos: "AutoAttak" in section 3, " It needs to emphasize here " -> " It needs to be emphasized here" in section 5
>
> Thank you very much.

---

> > ### Comment · Reviewer_qx2w · 2021-11-26
> > **Thank you for your response**
> >
> > Thank you for addressing my questions. Unfortunately, as with the other reviewers, I still have some concerns regarding the strength of the attack used to evaluate robustness. Nevertheless, I think this is a valuable contribution and I hope that future improved experiments can validate this defense.

---

### Official Review · Reviewer_9ZHX · 2021-11-04

**Correctness:** 3
**Technical Novelty And Significance:** 3
**Empirical Novelty And Significance:** 2
**Recommendation:** 6
**Confidence:** 4

**Main Review:**

### Strengths:
- The introduced defense is sound and novel
- The defense does not depend on adversarial training, so it is quite fast to train in practice compared to adversarial training.
- The defense is evaluated using standard white-box and blackbox attacks and compares to SOTA robustness benchmarks.

### Weaknesses:
- The new defense requires smoothening the input at test time, which requires multiple (200 - 400) forward-backward runs through the network, which might be impractical in some scenarios.
- The writing quality can be improved and it is hard to follow some sections on the paper.
- The submission does not include code.

### Detailed questions/suggestions:
- As I understand, at test time, the defense smoothens the input by adding noise and doing 200-400 iterations through the network to minimize the smoothing loss. Does this have to be sequential? Or could these be done in parallel (batches)? This matters for assessing the practicality of the defenses. That being said, it would also be helpful to add an inference time table to make this clearer for the reader.
- The paper mentions: “What is the consequence if we make the image smoother further, i.e., to decrease L_{F^l} ? The natural effect will be the increase of cross-entropy loss L_{ce} , but the drop in classification accuracy should be small since L_{F^l} and L_{ce} are comfortable with each other.” in an attempt to justify or motivate the introduced loss? But I didn’t really get what this means. What does it mean that two losses are “comfortable with each other”? I think this is too hand-wavy and I encourage the authors to fix this.
- I am curious how the defense scales to ImageNet.
- I am not sure what Fig 4 and 5 are adding to the paper. What should the reader be looking at in these figures? I think adding takeaways of these figs in text or caption should be helpful.

### Small changes:
In Alg 1, “s” should be “l”.



**Summary Of The Paper:**

The paper introduces a new empirical defense against adversarial examples. The new defense is competitive to SOTA adversarial training based defenses, but does not use adversarial training. Specifically, the paper proposes adding an additional loss to the standard classification loss which “smoothens” the feature maps of a specific layer in the model. At test time, random noise is added to each input, then the input is smoothened out (in an attempt to remove the adversary), by minimizing the Smoothening loss introduced earlier with respect to the input. The introduced defense is rigorously evaluated using SOTA attacks and shows competitive robust accuracies to adversarially-trained models on CIFAR-10 and CIFAR-100.


**Summary Of The Review:**

Overall I think the introduced defense is novel and sound. The new defense is faster at training time than adversarial training, but slower at inference time. It achieves slightly better accuracies than SOTA and is properly evaluated to the best of my knowledge on white-box and black-box attacks. I will give a score of “marginally above the acceptance threshold” since the paper quality can be improved, and since I have several questions which I think require clarification from the authors (see above).

---

> ### Author Response · Authors · 2021-11-12
> **Response to Reviewer 9ZHX**
>
> “As I understand, at test time, the defense smoothens the input by adding noise and doing 200-400 iterations”
>
> Sure, we do it in parallel. Since our network is fixed, all inputs are independent of each other, this parallelism can be easily achieved by pytorch. The “while” condition in 4 of algorithm 1 can be implemented with “torch.clamp” in the loss function. Of course, in a batch, some images may reach exit condition in the “while” loop earlier than others, which is fine,  as “torch.clamp”  will disable gradients of those images. So basically, the execution time depends on the images that need the most updates in the batch. Also, since we use the l=0 layer, which is closest to the inputs, it will facilitate the quick forward/backward propagation. The following table list a
> time comparison for normal inference and ours, with batch size=250, 40 batches in total. The experiment is based on PyTorch1.7.1, using NVIDIA GeForce RTX 3090.
>
> | Dataset | Normal Inference | Ours (total iterations) |
> | :----:| :----: | :----: |
> | CIFAR10 | 4.4888 s    | 1606.9329 s (13307)|
> | CIFAR100 |  4.5232 s    | 1406.0854 s (11602)  |
>
> “The paper mentions: “What is the consequence if…”
>
> Sure, you are right. There is some confusion about this wording. According to the common comment 2, let’s rephrase it this way.
> What is the consequence if we make the image smoother further, i.e., to decrease L_{F^l} ?. This essentially will remove some brittle features, however, since we train the network with L_{F^l}, the network somehow understands that the remaining features are still useful for image classification. In other words, the drop in classification accuracy should be small.
>
> “I am curious how the defense scales to ImageNet.”
>
> Yes, we are also excited to see the results, however, due to limited facilities, we can’t try that right now.
>
> “I am not sure what Fig 4 and 5 are adding to the paper. What should the reader be”
>
> Sorry about that. Here we try to give a visual impression of the active defense. From this, we can see that the feature smoothing distills the feature maps and reconstructs the sematic meaningful structures of the original inputs. We will surely add more descriptions to the paper.

---

> > ### Comment · Reviewer_9ZHX · 2021-11-26
> > **Thanks for the reply**
> >
> > I thank the authors for addressing my questions. I agree with the other reviewers about the importance of testing the robustness of the proposed method against adaptive attacks. Thus, I encourage the authors to conduct such experiments.

---

### Author Response · Authors · 2021-11-12
**Common Comments**

Dear Reviewers:
Thank you for all your reviews. As pointed by Reviewer 9ZHX, In Alg 1, “s” should be “l”. We are very sorry about any confusion that may arise, and special thanks to Reviewer 9ZHX!

These are some common comments we would like to emphasize.
1.To the best of our knowledge, besides the fast standard training time, our approach is the first method that can handle epsilon=8 and 16 simultaneously without changes of any hyperparameters. In other words, our method is ignorant of attack models, which is necessary for practical settings, and unfortunately can’t be satisfied by all other current methods, even the manifold method of (Jin & Rinard, 2020), they use epsilon in the regularizations. Although we use WRN-28-10, a light model, for epsilon=8 of CIFAR10, we still rank No.1 in AutoAttack; for epsilon=8 of CIFAR100, only slightly behind WideResNet-70-16. The most impressive thing is that other SOTAs drop significantly when epsilon=16, while ours still keeps high robust accuracy. For example, in table 2, (Rebuffi et al., 2021) only achieves 25.39%/12.45% for CIFAR10/CIFAR100, while ours can get 54.24%/22.76%.

2.The key motivation of our active defense is that we try to smooth out the adversarial noise in a sensible way. Let’s assume the network before the feature layer where we derive feature loss as N1 and the remaining one as N2. Basically, there are two conditions to be satisfied:
   (1) N1 acts as a low pass filter which is achieved by our feature smoothing loss.
   (2) The low pass features generated by N1 should be discriminative for N2, which is achieved by standard cross-entropy loss.

However, it is not enough. An adversary can always generate a malicious pattern that can pass through N1 and fool N2, since N1 is not a perfect low pass filter, and N2 can still accept brittle yet discriminative features. The active defense can actually remedy this by promoting feature uniformity following noise injection to the input.  This noise is a uniform random noise with sigma=0.075, which is about 19/255, far higher than 8/255. So the effects of adversarial noise should be destroyed to a big extent. And thanks to 2), after strong feature smoothing, the resulting image, which is about 25/255-32/255 in l-infinity norm from the clean one, still can be classified correctly with a high rate.

3.As regards obfuscated gradients, we totally understand this concern. People usually rely on the gradient to craft adversarial images. If it is hard to do this, then “obfuscated gradients” concern will naturally arise. However, according to the above common comment 2, our design is very different from conventional ones, we don’t exploit any gradient manipulation to enhance robustness. We also try our best to defeat this via adaptive attacks.
Here we can also provide another perspective. Defenses and attacks are always in a race, there is no point to suspect the strong defense which is hard to attack as ‘’obfuscated gradients” as if it breaks the rules. Being difficult to attack itself manifests the success of defenses.  We hope that as the research goes on, there should be less concern of “obfuscated gradients”.

4.we all adhere to reproducible research, and will release code at an appropriate time.

---

### Author Response · Authors · 2021-11-22
**Another High-Level Interpretation of Our Solution**

Dear Reviewers:

We provide another high-level interpretation of our solution. Again, we refer to the network before the feature smoothing loss as N1, and the remaining one as N2. In training, we only use N1+N2. However, in testing, when we unfold the while loop and refer to the gradient of N1 as N11, the network will be (N1+N11) + (N1+N11) + (N1+N11) + … + N1 + N2, where the exact number of repetitions of (N1+N11) depends on the input. To the best of our knowledge, this is the first run-time reconfigurable network for adversarial defense.

Reviewer tMiw raises a question regarding the adaptive attacks on (N1+N11) + (N1+N11) + (N1+N11) + … + N1 + N2. This question is very important, and we take it very seriously. However, we run into technical difficulties, since a long string of (N1+N11), whose length can be 200-400 for epsilon=8, and up to around 500 for epsilon=16, will pose a serious challenge to GPU memory when deriving the gradients. In fact, each defense is weak given enough computational resources. In other words, there is no point to evaluate the robustness without the constraints of resources. So, our approach is robust at least from this perspective.

Thank you very much!

---

> ### Comment · Reviewer_tMiw · 2021-11-22
> **Not convinced by your argument.**
>
> I agree that it may be computationally expensive to run a true adaptive attack on your system, especially against a version that performs many gradient steps, although you could try it against your defense with fewer gradient steps (by using higher beta).  However, I think the bigger point being made here is incorrect.  The point of trying adaptive attacks is not just to show that a defense is difficult to break, but we are actually interested in whether or not there exists an adversarial example inside the L-infinity ball of a particular radius surrounding the original sample.  This is why certifiable defenses exist and underpins why we work so hard to verify the robustness of empirical defenses as well.  If no such adversarial example exists, then there is no "race" between attacks and defenses.  While you did not break your own defense in your work (and I point out that you have an incentive to fail at this objective), breaking your defense might NOT be difficult for others, for example using the adaptive attack or perhaps a cheaper surrogate.  I will conclude by noting that previous work of Carlini has demonstrated that numerous proposed defenses which the original works did not try hard enough to break are easily defeated by simple attacks, and so I think the authors of this paper need to try proper adaptive attacks before this paper can be considered for publication.

---

> > ### Author Response · Authors · 2021-11-23
> > **Response to Reviewer tMiw**
> >
> > "The point of trying adaptive attacks is not just to show that a defense is difficult to break, but we are actually interested in whether or not there exists an adversarial example inside the L-infinity ball of a particular radius surrounding the original sample."
> >
> > You are talking about absolute security, which is important. However, we are interested in relative security with respect to computational complexity, which is more important in practice. For example,  according to Wikipedia, “The MD5 message-digest algorithm is a cryptographically broken but still widely used hash function producing a 128-bit hash value, …., may be preferred due to lower computational requirements than more recent Secure Hash Algorithms.”
> >
> > "I point out that you have an incentive to fail at this objective"
> >
> > NO. As stated in the paper “we try our best to defeat our defense scheme”, which is also recognized by Reviewer 9ZHX, we are always upholding high standards of scientific excellence. We appreciate all feedbacks from reviewers. The reason we don’t try EOT at the time of submission is that we think it is quite irrelevant since our method is very different from traditional ones.
> >
> > "…previous work of Carlini has demonstrated that numerous proposed defenses which the original works did not try hard enough to break are easily defeated by simple attacks…"
> >
> > True. However, we would like to provide another perspective. All current methods only deal with epsilon=8 attacks, and adaptive attacks have to be confined to epsilon=8, which is neither necessary nor reasonable. It reflects the embarrassments of the weak defense. Attacks with epsilon=16 can also be considered as adaptive ones, which is much simpler to be implemented and evaluated.
> >
> > Our approach can simultaneously deal with epsilon=8 and 16 with high performance as shown in the paper. If our paper is rejected due to insufficient evaluation, then others without testing epsilon=16 attacks should be desk rejected.

---

> > > ### Comment · Reviewer_tMiw · 2021-11-23
> > > **Still not convinced.**
> > >
> > > "Attacks with epsilon=16 can also be considered as adaptive ones."
> > > This is just false.  Adaptive attacks are not simply ones which are less constrained, they are ones which are finely tailored for defeating a particular defense, and you have not tried a true adaptive attack.  This is not simply about "insufficient evaluation", your paper is missing the single most important piece of evaluation.
> > >
> > > "If our paper is rejected due to insufficient evaluation, then others without testing epsilon=16 attacks should be desk rejected."
> > > No, you are again missing the point.  Since you did not perform true adaptive attacks, I'm not sure that your defense is robust w.r.t. epsilon=8 OR epsilon=16.
> > >
> > > "The reason we don’t try EOT at the time of submission is that we think it is quite irrelevant since our method is very different from traditional ones."
> > > Your defense uses randomization in a way that is perfectly appropriate for EOT.  Namely, you can find an attack that is successful on expectation over many realizations of randomness, exactly the same way people use EOT on any randomized defense.
> > >
> > > Finally, I don't think MD5 is particularly relevant here, but the same Wikipedia article you cite is against your argument.  Namely, it indicates that experts say MD5 should NOT be used:
> > > "As of 2010, the CMU Software Engineering Institute considers MD5 'cryptographically broken and unsuitable for further use'."
> > > Similarly, you have NO guarantee on the computational complexity of breaking your own defense.  Personally, I would accept your paper without this type of guarantee if you would only run a convincing adaptive attack, which you haven't.

---

> > > > ### Comment · Area_Chair_RPsY · 2021-11-23
> > > > **Thanks for the nice discussion**
> > > >
> > > > ... but please try to keep emotions out.
> > > >
> > > > I agree that adaptive attacks are essential for this paper. I also just found out that AutoAttack has new "automatic checks for potential cases where the standard version of AA might be non suitable or sufficient for robustness evaluation".
> > > >
> > > > Two of the listed cases are
> > > > - randomized defenses (they recommend to use the version "rand" - which I guess uses EOT)
> > > > - Optimization at inference time
> > > >
> > > > Both apply for the present paper. Therefore, in this case, the evaluation with AutoAttack could significantly overestimate the actual robustness and thus adaptive attacks are needed, which are tailored to the defense mechanism, see e.g. Carlini et al, On Evaluating Adversarial Robustness, https://arxiv.org/abs/1902.06705

---

> > > > > ### Comment · Reviewer_R7ZE · 2021-11-24
> > > > > **I agree with the excellent conclusion of Area Chair RPsY.**
> > > > >
> > > > > I can feel the author's intention to promote the development of adversarial learning, and I do agree with the authors that all of us in the adversarial learning community hope to improve the robustness of DNNs, but, at the same time, we should avoid the false robustness, which may hinder the development of adversarial learning. In this regard, AC pointed out two cases that may cause false robustness, i.e., randomized defenses and optimization at inference time. Thus, I suggest re-exploiting the effectiveness of the proposed method where these operations are removed.

---

### Decision · Program_Chairs · 2022-01-20

**Decision:**

Reject

**Comment:**

This paper suggests a novel defense against adversarial perturbations where during training a loss term is added which enforces similar feature representations.
At test time: i) noise is added, ii) the feature loss is minimized

The authors report excellent results against AutoAttack but the problem is that AutoAttack expects a static, non-randomized defense. Both is not the case for the defense proposed in the present paper. Therefore,  the evaluation with AutoAttack could significantly overestimate the actual robustness and the evaluation of the paper is therefore not valid. Thus adaptive attacks are needed, which are tailored to the defense mechanism, see e.g. Carlini et al, On Evaluating Adversarial Robustness, https://arxiv.org/abs/1902.0670.

As two reviewers noticed, the suggested "adaptive attack" in the paper is not properly attacking the whole defense mechanism by unrolling the test time optimization and using additionally EOT. Thus it is unclear at the moment if the method is really robust. Moreover, the inference time is significantly increased so that it is questionable if this approach is practically relevant. Therefore this paper is not ready for publication yet.